# Human–Machine Integration in Processes within Industry 4.0 Management

**DOI:** 10.3390/s21175928

**Published:** 2021-09-03

**Authors:** Javier Villalba-Diez, Joaquín Ordieres-Meré

**Affiliations:** 1Hochschule Heilbronn, Fakultät Management und Vertrieb, Campus Schwäbisch Hall, 74523 Schwäbisch Hall, Germany; javier.villalba-diez@hs-heilbronn.de; 2Department of Artificial Intelligence, Escuela Técnica Superior de Ingenieros Informáticos, Universidad Politécnica de Madrid, Boadilla del Monte, 28660 Madrid, Spain; 3Escuela Técnica Superior de Ingenieros Industriales (ETSII), Universidad Politécnica de Madrid, 28006 Madrid, Spain

**Keywords:** Industry 4.0, Operator 4.0, process variability, JIDOKA, integration explaining variability

## Abstract

The aim of this work is to use IIoT technology and advanced data processing to promote integration strategies between these elements to achieve a better understanding of the processing of information and thus increase the integrability of the human–machine binomial, enabling appropriate management strategies. Therefore, the major objective of this paper is to evaluate how human–machine integration helps to explain the variability associated with value creation processes. It will be carried out through an action research methodology in two different case studies covering different sectors and having different complexity levels. By covering cases from different sectors and involving different value stream architectures, with different levels of human influence and organisational requirements, it will be possible to assess the transparency increases reached as well as the benefits of analysing processes with higher level of integration between them.

## 1. Introduction

Value chains associated with Industry 4.0 (I4.0) are formed by complex cyber-physical networks in which humans and machines process information efficiently to supply a customer with the desired product [1,2,3]. I4.0 and industrial internet of things (IIoT) describe new paradigms for integrated human–machine interaction [4,5]. Both concepts are based on intelligent, interconnected cyber-physical production systems that are capable of controlling the process flow of industrial production. Since machines autonomously make many decisions and interact with production planning and manufacturing systems, the integration of human users requires new paradigms [6].

IIoT technology is significantly contributing to enlarge the data available for many manufacturing processes. In an I4.0 context, such as the IIoT [7,8], these data are produced by decentralized sources such as thousands of sensors in factories [9], i.e., the data are distributed over networks [10]. With the classical already collected dataset related to sensors located at the processing machines, now it becomes possible considering additional data coming from wearables of human operators [11]. The number of edge devices that are currently developed to support fitness and health monitoring is enormous [12]. Many of them aim at measuring body parameters to offer care related services [13]. At the same time, a lot of smart health applications are developed, often making decisions or offering feedback based on sensor data processing. Application developers often struggle to integrate and plug in novel sensor technologies, becoming available on the market at a fast pace [14]. These technologies enable describing processes in a more integrated way, including many more potential sources of variability [8]. Although the advantages are rather evident, still there are significant challenges to be better identified and faced when new useful solutions regarding knowledge and management are foreseen.

In the manufacturing IIoT I4.0 domain, the I4.0 vision has promoted smart manufacturing and smart factory concepts by augmenting all assets with sensor-based connectivity [15]. These intelligent sensors generate a large volume of industrial data helping to create digital twins (defined as a digital replication of both living and inanimate entities that enable seamless data transfer between the physical and virtual worlds) as support for a live mirror of physical processes [16,17]. Within this approach, the ambition is to capture the process variability, being able to process all relevant information by big data analysis on cloud computing so that manufacturers are able to find manufacturing processes’ bottlenecks, identify the causes and impacts of problems in such a way that effective implementation of measures becomes useful either for product design or for manufacturing engineering including maintenance, repair and overhaul [18].

A critical aspect to be considered when the previous interest is addressed is the human influence on the processes. There is a gap between the information collected by the IIoT devices and their capability to capture the causes influencing process variations. This human–machine symbiosis presents great potential advantages, since on the one hand the human has a great cognitive flexibility that the machine lacks, while on the other hand the machine has a great computational capacity superior to the human [19]. However, there are also voices warning of the potential dangers of the *bionization* of human tasks [20,21]. Fundamental requirements for the future design of human–machine interactions in productive assembly systems are now being identified [22]. Expectations generated by Operator 4.0 (O4.0) in this context have implications for empowerment and management models [23]. The technical implications of realising a human–machine symbiosis have to enable the use of trustworthy and ethical artificial intelligence (ethical AI) [24].

The aim of this work is to use IIoT technology and advanced data processing to promote integration strategies between these elements to achieve a better understanding of the processing of information and thus increase the integrability of the human–machine binomial, providing appropriate management strategies for these configurations [25,26,27]. Thus, the major objective of this paper is to evaluate how human–machine integration helps to explain the variability associated with value creation processes. Therefore, the research question being addressed in this paper can be formulated as *RQ1: The I4.0 technology allows to increase the transparency to understand process variability when it is used to integrate different sources of uncertainty.*

In particular, we are interested in the case of natural intereffects between human workers and operating machinery. The approach selected is to implement an action research methodology through two different case studies covering different sectors and having different complexity levels, and the presentation is structured into four further major sections. First, in Section 2, we outline the main lines of research that deal with the human–machine integration in an I4.0 environment. Second, in Section 3 we present the results of two case studies that illustrate the usage of IIoT technology when integrating the human–machine binomial. Third, in Section 4 we discuss the possible implications for the management of creating processes. Finally, in Section 5 we present the conclusions, further steps, and possible limitations of this work.

## 2. State of the Art

Strategic organizational design is a scientific field [28,29] that studies the relationship between organizational entities and how its structure and functionality affects its performance [30]. Under the organizational network paradigm, modern organizations can be understood as a symbiotic socio–technical ecosystem of social networks [31] that interact with ever increasingly complex networked cyber-physical distributed interconnected sensors [32], whose readings are modelled as time-dependent signals on the vertices, human or cyber–physical, respectively.

Under this evolutionary information flow perspective [33], organisations can be adequately modelled [34]. One of these models is the Human–Cyber–Physical Systems (H–CPS) model, that integrates the operators into flexible and multi-purpose manufacturing processes. The primary enabling factor of the resultant O4.0 paradigm is the integration of advanced sensor and actuator technologies and communications solutions. Although process automation reduces costs and improves productivity, human operators are still essential elements of manufacturing systems [35]. As discussed in [36], the degree of automation does not directly imply an enhanced operator performance, because handling human factors requires more complex dimensions related to human-to-machine interactions, including robotics. The integration of workers into an I4.0 system consisting of different skills, educational levels, and cultural backgrounds is a significant challenge. The new concept of O4.0 was created for the integrated analysis of these challenges [19].

The concept of O4.0 is based on the so-called H–CPSs designed to facilitate cooperation between humans and machines [36]. Although specific contributions regarding different dimensions have been proposed by different authors [23,37] still there is a significant room for improvement when an integrated perspective is required, because the available wearable devices lack of enough level of integration. In current industrial practice, most applications are developed in isolated circumstances aimed at addressing specific problems. Therefore, there is a gap in creating human-centred systems able to promote an operator learning context not only relying on single parameters but also providing a meaningful articulated set of relevant parameters both in the short and long term [37].

I4.0 envisions a future of networked production where interconnected machines and business processes running in the cloud will communicate with one another to optimise production and enable more efficient and sustainable individualised/mass manufacturing. Inside such a vision, there are different requirements to be considered, including cloud computing, data pseudo-anonymisation, as well as data micro-services. The shop-floor in virtual space is the reconstruction and digital mapping of the physical devices at shop-floor level. They exchange data/information/knowledge through by using a big data storage and management platform.

These shop-floor-management platforms construct a virtual shop-floor system that monitors the working progress and working status of assembly stations, products, and manufacturing resources in the physical shop-floor so that it can be dynamically, realistically, and accurately mapped in the virtual space through cloud services [38]. The main challenge to developing shop-floor in virtual spaces is addressed is the complexity of the IIoT solutions, as they suffer from poor scalability, extensibility, and maintainability [39,40]. In response to those challenges, microservice architecture has been introduced in the field of IIoT application, due to its flexibility [41], lightweight [42] and loose coupling [43].

The evolution of the human–machine integration that allows benefiting from the different information processing capabilities of both parties has been investigated in this context in a comprehensive manner [5]. Extensive and intensive research has shown that on the one hand, humans in shop-floor management environments in I4.0, have a holistic problem-solving capability where several brain areas are activated for problem solving [44,45,46], however humans have a limit to the cognitive load they can compute that affects their performance [47,48]. On the other hand, with the advent of artificial intelligence, machines are increasingly capable of performing a massive processing of information that can make up for human deficiencies: one approach is to use the machine, having greater computational capacity to reduce the cognitive load of humans [49,50,51], another approach has been to create a semantic framework that allows for machine recommendations for human problem-solving related to manufacturing tasks [52,53], while other scholars have proposed an open source web-based protocol to enhance inter-operability between human and machine assets [54]. The problem with all these proposals can be summarized in the fact that they try to adapt either the machine to man or vice versa, and as a natural consequence, they obviate a symbiosis between both forms of information computation.

Although previous research studies have addressed aspects related to human–machine integration, they were performed mainly from a dominant perspective, including process development design [55], but also analysing the relationship between management practices and Industry 4.0 as in [56]. However, a low number of contributions were focused on how a more integrated view provided with the help of I4.0 allows to better understand causes of process variability, and this paper aims to contribute to reduce this gap.

## 3. Case Studies

As a first step to evaluate the effect of human–machine integration in I4.0 environments, two case studies are used. As argued by Byrd and Turner [57], a single case study can be seen as the only possible building block in the process of developing the validity and reliability of the proposed hypothesis. Following the recommendations of Eisenhardt [58], a clear case study road-map is followed for each one of them. This road-map has several phases:Scope establishment;Specification of population and sampling;Data collection;Standardisation procedure;Data analysis.

### 3.1. Case Study 1. Reverse Logistics Process. near Field Communication

In this case study, we focus on studying the variability experienced by a reverse logistics collection process of steel scrap, in which a human driver in a truck covers different routes and time periods. In particular, we concentrate on merging relevant and difficult to evaluate aspects such as the state of the drivers, their operational working conditions, and other health-related parameters with data related to the machine elements used to perform the logistics tasks.

#### 3.1.1. Scope Establishment

The ambition in this case is to merge technologies covering different aspects as a way to better define the influence of different factors. We placed these devices on the human worker and we considered that they do not affect their work, which can be performed normally.

Integration of data flows is relevant to produce process-related information, making it possible to extract behavioural rules.

#### 3.1.2. Specification of Population and Sampling

In this case study, we analysed the data of 5 users sharing 3 trucks, performing daily routes in 3 shifts. We monitored the data are monitored on a per second basis, but aggregated them by day to ensure a more consistent analysis.

#### 3.1.3. Data Collection

The initial goal is to assess the technologies themselves in a real case implementation, which include, as shown in Figure 1:Health-related parameters are gathered through non-invasive Bluetooth Low Energy (BLE) devices. In this study, we consider as irrelevant the effect that the fact that their health is being measured could have on human behaviour.Trucks’ condition monitoring is gathered through solid state based devices.Near-field communication (NFC) Technologies.

Specific Android based applications have been developed to enable data collection and sensor fusion [59], as well as the integration of the NFC tags with the process logic in a consistent way [60] through a mobile phone.

Specifically, from MongoDB [61], we have created a platform for storing vehicle driver wristband data. They are organized by date (day) and MAC identification of the wristband. We have also created a read user in the MariaDB manager cluster system, a community-developed fork of the MySQL relational database management system [62], which stores lower frequency data from different sensors and web services. Every few minutes they run processes that load data into their databases which can be accessed with a certain user and password.

#### 3.1.4. Standardization Procedure

The key performance indicators (KPIs) measured are standardized over the whole management system to ensure a comparable framework in which several processes can be benchmarked against each other. A list of the measured KPIs and its meaning is depicted in Table 1.

#### 3.1.5. Data Analysis

To perform the analysis of the KPIs, a preliminary analysis of the coefficients of variation (CV), as the ratio between the standard deviation and the mean of the KPI. It becomes interesting to see how the CV is much more sensitive to the rather distant multimodal structure. To avoid such effects, the coefficient interquartile of dispersion (CQD) was introduced. Figure 2 shows that some KPIs have the largest CQD and these KPIs are good advisors for the variability of processes. Therefore, attention will be given to the reasons for such variability and to do this a paired view of the KPIs is presented in Figure 3.

To assess the variability in more detail, it was decided to analyse records obtained because of the daily activity, by assigning to each variable its quartile class (see Table 2), additionally class-oriented variables such as Shift, TruckPlate and DriverID were included to represent all relevant elements potentially linked to process variability (see Table 3). In this way, we transform the records of KPIs related to the same operation cycle in terms of an orchestrated list of KPIs quartiles that can be then extended to all operations, creating a sort of item list. The goal is to apply data mining (DM) to obtain potentially useful, previously unknown, and ultimately understandable knowledge from the data. Association rule mining is one of the important portions of data mining and is used to find interesting associations or correlation relationships between item sets in mass data (item list) [63].

To apply the DM association rule technology, the selected algorithm was FP-Growth (frequent-pattern growth), which is an improved algorithm of the Apriori algorithm put forward by [64]. It compresses data sets to a FP-tree, scans the database twice, does not produce the candidate item sets in mining process, and greatly improves the mining efficiency [65].

After creating the item list, mining for rules having limited support but high confidence can start. Rules do not extract an individual’s preference, rather find relationships between the set of elements of every distinct transaction. This is what makes them different from collaborative filtering. Normally, rules exhibiting a high level of support are the so-called ’well-known rules’ the people involved in such activities are familiar with, but those having low support, although their confidence becomes even higher, are harder to learn and it is where DM can help to unveil those unknown behaviours.

In our application case, the threshold for support was established at 15% and the confidence threshold was established at over 95%. As a significant variability in CQD appeared regarding *kpi_tot_cycle*, which actually reflects all operations, it could be interesting to look for explanations with its highest range (kpi_tot_cycle_Q3) and the lowest one (kpi_tot_Cycle_Q0).

The relevant factors and combinations can be better understood, and filtering the right hand side (RHS) to contain *kpi_tot_cycle_Q3* it is possible to find what Table 4 presents. Similarly, lower values for the same KPI have been analysed with the same technology, where the findings are presented in Table 5.

Based on the more than four hundred items in the item list, more than 250000 rules have been distilled when the minimum support is chosen to one percent. Analysis of the selected RHS rules presented in Table 4 and Table 5 show several interesting aspects, such as that there are rules discovered that sometimes are meaningless from the practical business point of view, as in the case of Table 5, because the second rule assumes that the time spent by the customer during the collection of materials is a consequence, and it is part of the process as an antecedent and never a consequence. It also happens in Table 4 with rules 15, 16, 18, and 22.

Another relevant aspect is that the discovered rules establish a relationship between the total cycle KPI and the input time KPI as well as with the time used to collect materials at the customer’s site. This is interesting because those two KPIs were the most sensitive to CQD in Figure 2.

### 3.2. Case Study 2. Integration between Robotics and Human Oriented Processes

In this case study, there are three different processes involved, each of them with different levels of automation and human operator engagement. In Figure 4, the sub-process design is presented, where the manual forming of different components are manufactured and assembled in six different configurations (Rework station). After assembling, robotic laser-based quality control stations (SCAN units) verify the geometric tolerances of each part and when they do not pass the quality checks, they are routed back for manual repairing, getting integrated again for checking afterwards. After successful robotic inspection, a set of three pairs of manual inspection stations are configured to finally assess the parts (CHECK units) and attach the individual report before packaging and delivering products to customers (Final Gate Storage).

Here there are different sources of variability as per part reference. First of all, since part manufacturing involves a relevant amount of manual work and component integration, including robotic welding stations, the number of goods per time unit has some uncertainty. The second source of variability is because of the robotised inspection, as the quality criteria are rather ambitious, because the success ratio is not constant and requires significant reworking and reinspection activities. The last and most visible impact is for CHECK units, where the shortage of parts after a geometrical check on SCAN units damages the whole performance. Shortage of parts to be processed at CHECK stations hinders the productivity of these workers, while a type of rigid planning is imposed because of labour regulations and the required union assessment before adoption. Therefore, since it is not possible to dynamically allocate workers to different working places, then the management reaction is to protect the CHECK capacity increasing the intermediate buffer, which is against the lean philosophy and it complicates the shop-floor layout.

#### 3.2.1. Scope Establishment

To illustrate the significance of the issues captured by this study case Figure 5 presents the variability, where neither the finally delivered number of parts nor deviations from what it was initially planned are regular enough.

The strong variability found per day and shift looks interesting as there are robotic operations involved (SCAN units) (see Figure 4 which should add regularity because the more predictable cycle time values). Indeed, due to the labour regulations enforced in the country where such a facility operates, such variability (uncertainty) forces to allocate resources that sometimes are not able to perform as expected, which compromises the business dimension of the activity as a whole, either because of insufficient production or lack of productivity.

#### 3.2.2. Specification of Population and Sampling

To carry out a meaningful analysis, different datasets were collected inside individual processes daily based either on the automation system itself (SCAN units) or by slicing the time in a range of 30 min for manual operation of CHECK units. After several months of data, conclusions can be given in a clearer way. It was accepted that automation of the laser measuring system (SCAN units) continuously ingests products without delay, except those legal stops that are allowed for production, such as lunch time, which are well established. Therefore, more than 90,000 part components have been analysed, as they have been identified as the reference entity.

#### 3.2.3. Data Collection

Two different data streams have been considered as an example for the process-oriented analysis involving both automatic and manual operations, which require more integration and better understanding from the managerial point of view.

The first data stream is related to robotised inspection workplaces, where based on the previous hypothesis it is possible to estimate the most frequent time duration for inspecting each reference successfully, and based on it, to estimate production losses at such stations, making it clear that earlier or later, such production losses will impact the final manual inspection units.

The second one is coming from manual processes and to perform a consistency analysis, fixed time slots where defined and production per slot was measured. Time ranges of 30 min where analysed, looking to identify where production losses occurred. Since every inspection is around three minutes including handling and shipping, that means nine items per every half hour. Therefore, such a ratio was considered the gold standard.

#### 3.2.4. Standardisation Procedure

Standardisation plays an essential role here as it enables to define what the expectations for production are, and how big deviations appear.

By observing the delivery time of the individual parts, measured from the previous successful inspection, we can identify the performance losses occurring when the current part lasts for longer than expected as the most frequent value found for that part reference. This information is presented in Figure 6 just for a few references. Similar behaviour was identified for all different references regularly produced, and based on the findings, conclusions can be derived with implications at the managerial level.

Indeed, when standardisation covers several processes at once, additional and not previously detected sources of errors appear, mainly because the analyses were carried out by individual process units, therefore, the derived impacts become hidden. It is related to references not matching the naming rules because of spelling issues, or different labelling rules for different working units. Such lack of integrity along the value chain does not help to provide a comprehensive perspective of the whole process and this is because standardisation is so relevant during the structured analysis.

#### 3.2.5. Data Analysis

Real evidences show a high impact in the performance of the whole set of processes, where the nonproductive time in the last process exhibits significant variability depending on the process but also depending on the shift, as depicted in Figure 7.

For the robotised inspection units, losses can be also estimated, as depicted in Figure 8.

A relevant aspect evidencing that the shortage issues observed at CHECK units are actually due to the previous production steps can be observed from Figure 6, and it can be observed equally for any of the more than fifteen different part references manufactured at the shop-floor. It is derived from the histogram used to identify the effective time required for a reference to be successfully processed.

If the part reference REF01 is considered, it becomes clear that in most of the cases successful processing of this reference at the SCAN units lasts for 33 s. However, it is possible to realise that there are a relevant number of cases where it lasts for a shorter time, and in some cases it takes for longer. The main reason for shorter times is because when the robot decides the part fails, it does not continue to explore all positions and it rejects such a part, expending a shorter time than when the part is correct. On the opposite side, there are a significant number of parts lasting for 50 and 60 s, which means that after the last part successfully processed, there were several others with the same reference failing in fulfilling the QC requirements, therefore, after a while another part was OK but it took much longer than 33 s. From this situation, it becomes evident that SCAN units are testing a very relevant number of parts requiring reworking and demanding inspection several times. The direct effect of loosing efficiency is that the situation hinders process stability for the next production unit.

From the comparative analysis between SCAN and CHECK inspection units, it becomes clear how mitigation strategies such as enlarging part buffers between units have kept the situation bounded but with a high level of variability, but the effectiveness of SCAN continues to get degraded over time, mainly because managers are much more focused on the more evident problem of part shortage.

It is also worth to mention the different behaviour for the two robotic inspection lines presented in Figure 8, where SL01 is more stable and SL02 shows that it is not under control neither for the morning shift nor for the night one.

## 4. Discussion

When the first case study is considered, the integration of data from different sources, including routes, position, indoor and outdoor climate, NFC data provides accountability for different process steps. The integration was carried out because of the I4.0 principles and it allows to collect evidences capable of explaining process variability with increased confidence than ever before. Just to illustrate such reality, before deploying this project, managers in the company were explaining the variability in process duration because of weather conditions or because of driver attitudes, and in some cases because of route congestion. Clearly, they based their analyses mainly on beliefs or personal experiences, but after the analysis carried out, it becomes obvious that the main sources of variability are not because of those reasons but because of bottlenecks in the input weighting system at the customer’s site. With lower frequency but yet importantly, the time spent inside the customer site is also significant, but it was not the particular shift, the truck or the driver involved.

The deployed technology and analysis capability can be easily repeated on a regular basis, trying to inform managers about the effectiveness of the adopted measures, but also allowing to look for different objectives (RHS filtering), or even to predict the duration of specific process steps based on different parameters.

Regarding the second case study, it is needed to recognise that the processing logic is here rather complex because the components come from different manufacturers and at the shop-floor there are different stations involved in the whole Value Stream Mapping (VSM) producing different additional components, all of them welded and assembled at other production units, where the final quality control (QC) is done, both on SCAN units and then on CHECK ones. Because of such nonlinear circuit, since when the part is rejected due to QC reasons, it is extracted, reworked accordingly, and resubmitted, as well as because of the previously explained complex process with material flow through a large amount of steps and the parts becoming mixed between stations and part variants, it is very hard to track the different causes for failures. Therefore, the classical applications of Pareto techniques to identify main sources of failures do not work.

Sources of failure can be generally related to different root causes including workers in previous production stations, but also to the process and inspection systems, having more than one hundred of those identified causes, non-regularly distributed over time or part reference.

In this complex environment, only the I4.0 approach becomes effective, as the system collects information from workers’ performance per working place, the period of time, and references produced, while additional postprocessing is needed because sometimes the number of nonconformity events per part is much larger than expected, for instance additional bending of the part can negatively impact on different distances and angles between points. It also collects information from the automatic scanning system and from workers’ wearable elements, helping to complement the process perspective.

The advantage of the adopted approach as described in this case is that it increases the transparency of the processes as well as the effectiveness of the managerial decisions made. Instead of relaying on impressions from different people which, although very skilled, also have their own biases, the tools provide an agnostic perspective on the process and their impact.

Just as a small example, it was found that as opposite to the initial thoughts suggested by the internal experts, worker participation at CHECK units looked to moderate the impacts of uncertainties and part shortages. This is because different strategies can be promoted looking to minimise wasted extra capacity in specific periods, while automatic systems are much more rigid and require technical interventions lasting for a longer period of time. Therefore, human contribution provides significant flexibility to the processes, although the lack of production is still there.

The interest of the company now is to better integrate previous production steps, which in the beginning seemed rather independent, but new technologies are required to get them much more integrated into the analysis, continuing increasing the transparency.

## 5. Conclusions

This paper tries to highlight the significant contribution of the I4.0 framework to hybrid processes, where automated but also manual processes are required to cooperate and where most of the management strategies based on the ’split and win’ approach fail to provide consistent evidences to improve the business.

Major contribution is related to increasing transparency from an agnostic perspective, avoiding bias because of beliefs or other reasons. Another significant contribution is the integrative perspective brought, as it enables rather easily to go through a very high number of items, being processed in different ways and different places and in different time periods, connecting all points much more consistently than even before. In addition, a hidden benefit is connected with the incremental characteristic of the analyses because of the different requests raised by the managerial aspects can be better addressed with the additional information provided by means of the I4.0 and the integration of process-oriented data streams. This is relevant because providing answers to a managerial question takes the interest towards the next one, which connects with scalability of the research.

A relevant aspect highlighted by this research is that a consistent, deep, and transparent analysis can be carried out, still protecting workers’ identities by avoiding explicitly placing the focus on them, as required by the ethical AI principles.

It is needed to recognise that this paper is not claiming that the application of extended I4.0 technology to process-oriented integrated data flows will get the same level of benefits, because it will be case dependent (process and management). By exploring the presented two cases, the evident limitation related to the scarcity of the sample appears clearly. However, as they cover different business units from different sectors (logistics and production), with different complexity levels, and they brought abstract properties increasing the existing prior knowledge about the reasons for process variability, we believe they can be applied in other cases as well, with a similar increase in the existing knowledge. Based on these facts, this paper found enough support to positively answer the identified research question.

In terms of future activities, since the interests of companies are business driven, but at different speed. In the first case, they are interested in developing forecasting models and learning about their robustness able to better schedule the logistics activities, while in the second one they are just focused in increasing the understanding of the complex production process they are managing. In this particular case, they are aligned with requirements from the I4.0 paradigm, which is rather pervasive and to check influences from any source, looks to ingest significant behaviours from all relevant shop-floor units. Indeed, to complete the interaction requirements tracking, some wearable devices are under consideration, since they can help to check process variability and related effort from workers.

Finally, a common request is to bring a convenient way to present relevant information to managers in an automatic way and rule driven. To this end, the current implementation uses light clients with plotly and trello tools but probably they will come up with additional requirements in line with process evolution as well as the need for improvements based on the decisions made. Therefore, integrating forecasting capabilities as well as a more integrated way to describe the VSM are under further investigation. All of them look to contribute to the health of the VSM, as an extension of the well-known concept of assets’ health.

## Figures and Tables

**Figure 1 sensors-21-05928-f001:**
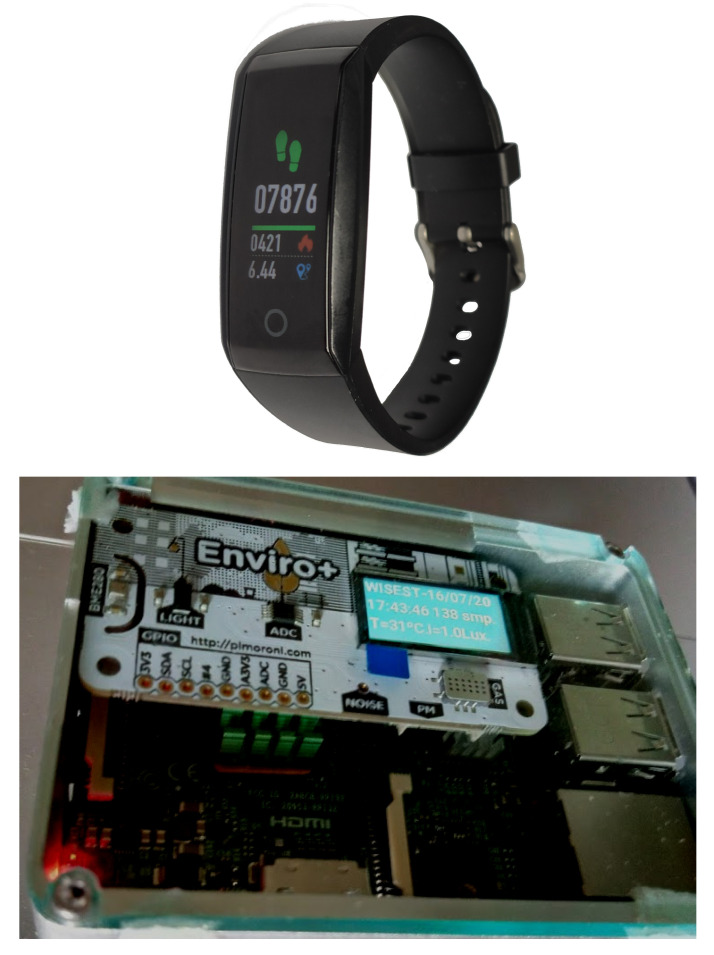
Some devices implemented in the case study 1. Smartbands to collect stress levels on top subfigure and Single Board Computers with appropriate plugin sensors on bottom subfigure.

**Figure 2 sensors-21-05928-f002:**
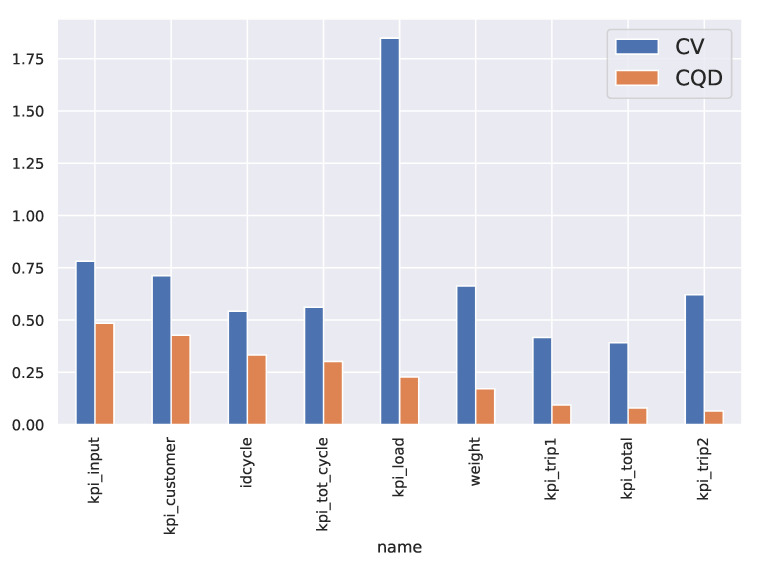
CV and CQD per variable in case study 1. Units for CQD are same that for original variables (see Table 1).

**Figure 3 sensors-21-05928-f003:**
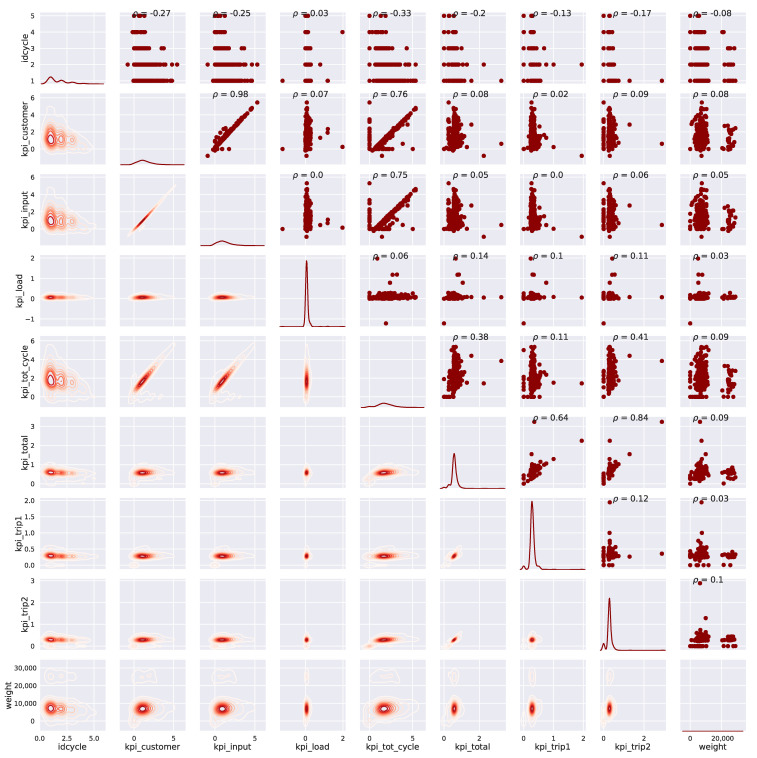
Correlation between KPIs in case study 1. Units for original variables were defined at Table 1.

**Figure 4 sensors-21-05928-f004:**
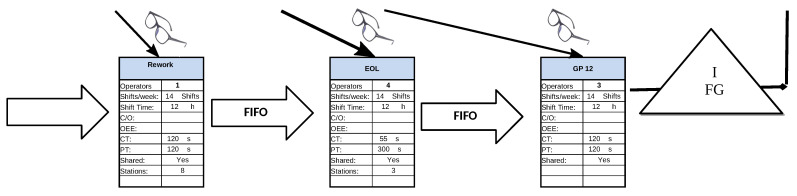
Value Stream Design (VSD).

**Figure 5 sensors-21-05928-f005:**
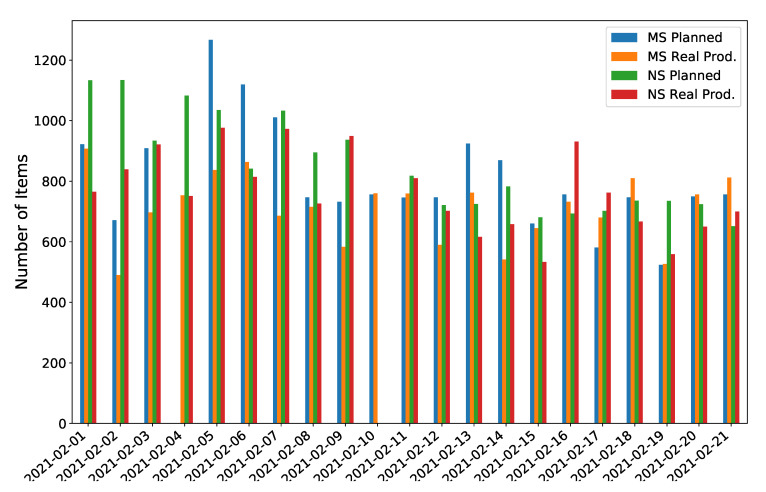
Variability between planned and delivered parts per shift: Morning (MS) and Night Shifts (NS) during February 2021.

**Figure 6 sensors-21-05928-f006:**
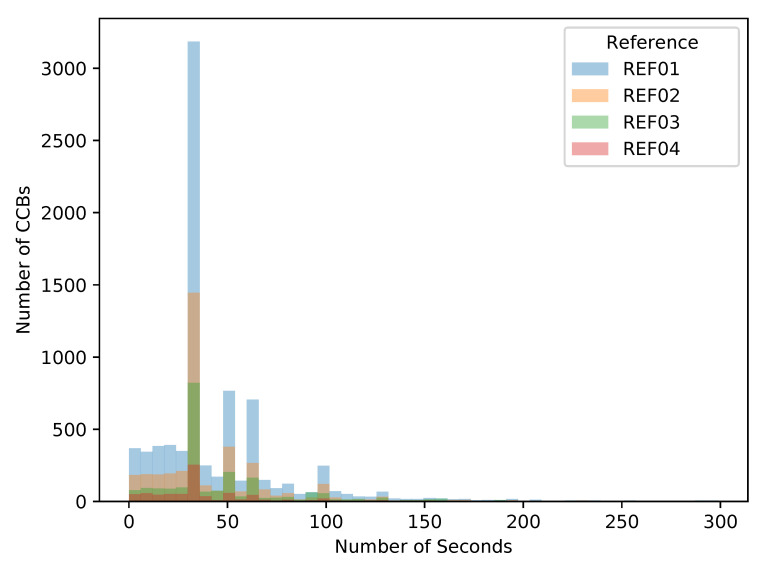
Standardisation for process duration at robotised inspection depending on the reference.

**Figure 7 sensors-21-05928-f007:**
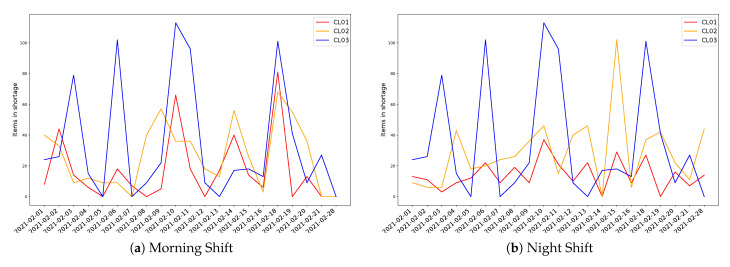
Production Losses at last inspection station per production line and Shift.

**Figure 8 sensors-21-05928-f008:**
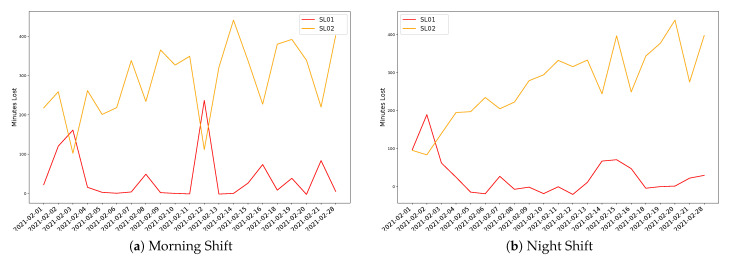
Production Losses at SCAN inspection units per production line and Shift.

**Table 1 sensors-21-05928-t001:** Monitored KPIs Case Study 1.

Name	Meaning
date	date for the record.
shift	Number of shift. 1: 06:00–14:00; 2: 14:00–22:00; 3: 22:00–06:00+1
plate	Truck plate ID.
user	Anonymous user ID (pseudo-anonymity for the truck driver).
idcycle	Number of cycle in the working day.
kpi_unload	Duration in minutes to unload the truck at the headquarters.
kpi_trip1	Duration in minutes from headquarters to customer facilities for collecting the scrap.
kpi_customer	Duration in minutes inside the customer facilities.
kpi_input	From customer entrance to loading point.
kpi_output	From loading point to the exit.
kpi_load	Duration of scrap loading process.
kpi_trip2	Duration in minutes from customer facilities for collecting the scrap to headquarters.
kpi_total	Duration of the whole cycle without headquarters movements.
kpi_tot_cycle	Duration of the total cycle.
weight	Scrap weight.
t2	Absolute time for starting the cycle.

**Table 2 sensors-21-05928-t002:** Quartile ranges for the interesting KPIs.

KPI (Unit)	Min	Q1-Init	Q2-Init	Q3-Init	Max	StDev
idcycle (h)	1.000000	1.000000	1.000000	2.000000	5.00000	0.945672
kpi_customer (h)	−0.801667	0.730278	1.187360	1.821110	5.44750	0.961633
kpi_input (h)	−0.905833	0.574653	1.036945	1.656458	5.29806	0.940865
kpi_load (h)	−1.217780	0.050208	0.063750	0.079792	1.97444	0.151037
kpi_tot_cycle (h)	0.000000	1.283052	1.745555	2.393260	5.34778	1.047631
kpi_total (h)	0.000000	0.534722	0.571111	0.626667	3.23611	0.228157
kpi_trip1 (h)	0.000000	0.259653	0.281805	0.313402	1.94278	0.123308
kpi_trip2 (h)	0.000000	0.267500	0.284722	0.304791	2.88056	0.177552
weight (Kg)	0.000000	5830.000000	6960.000000	8245.000000	28,780.00000	5271.877027

**Table 3 sensors-21-05928-t003:** Itemlist from the process records to be used for rule construction.

ItemListID	ItemList
1	(Plate_01, U_1, Shift_Q1, idcycle_Q0, kpi_customer_Q0, kpi_input_Q0, kpi_load_Q0,
	kpi_tot_cycle_Q0, kpi_total_Q0, kpi_trip1_Q1, kpi_trip2_Q0, weight_Q0)
2	(Plate_02, U_4, Shift_Q1, idcycle_Q0, kpi_customer_Q0, kpi_input_Q0, kpi_load_Q0,
	kpi_tot_cycle_Q0, kpi_total_Q3, kpi_trip1_Q3, kpi_trip2_Q3, weight_Q3)
...	...

**Table 4 sensors-21-05928-t004:** Rules explaining the Q3 for the *kpi_tot_cycle, where* ^*means logical and*.

	Antecedent_STR	Consequent_STR	Confidence
9	kpi_customer_Q3^kpi_total_Q3	kpi_tot_cycle_Q3	1.000000
10	kpi_input_Q3^kpi_total_Q3	kpi_tot_cycle_Q3	1.000000
15	kpi_input_Q3^kpi_total_Q2	idcycle_Q0^kpi_customer_Q3^kpi_tot_cycle_Q3	0.967742
16	kpi_customer_Q3^kpi_input_Q3^kpi_total_Q2	idcycle_Q0^kpi_tot_cycle_Q3	1.000000
18	idcycle_Q0^kpi_input_Q3^kpi_total_Q2	kpi_customer_Q3^kpi_tot_cycle_Q3	0.967742
22	kpi_customer_Q3^kpi_total_Q2	idcycle_Q0^kpi_tot_cycle_Q3	1.000000
24	idcycle_Q0^kpi_customer_Q3^kpi_total_Q2	kpi_tot_cycle_Q3	1.000000
58	kpi_customer_Q3^kpi_input_Q3^kpi_trip2_Q3	kpi_tot_cycle_Q3	1.000000
60	idcycle_Q0^kpi_customer_Q3^kpi_trip2_Q3	kpi_tot_cycle_Q3	1.000000
62	idcycle_Q0^kpi_input_Q3^kpi_trip2_Q3	kpi_tot_cycle_Q3	0.975000

**Table 5 sensors-21-05928-t005:** Rules explaining the lower quartile values for the *kpi_tot_cycle*.

	Antecedent_STR	Consequent_STR	Confidence
1	idcycle_Q1^kpi_customer_Q0	kpi_tot_cycle_Q0	0.972973
2	idcycle_Q1^kpi_input_Q0	kpi_customer_Q0^kpi_tot_cycle_Q0	0.972222
3	idcycle_Q1^kpi_customer_Q0^kpi_input_Q0	kpi_tot_cycle_Q0	0.972222

## Data Availability

The data presented in this study are available on request from the corresponding author. The data are not publicly available due to restrictions from the companies involved in the case studies.

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
