# Peer review of "Human–Machine Integration in Processes within Industry 4.0 Management"

_sensors, 2021, doi:10.3390/s21175928_

Round 1
Reviewer 1 Report
The manuscript submitted for review, titled: "Human–Machine integration in processes within Industry 4.0 Management" is very interesting.
To improve it, here are my specific comments:
- Abstract needs improvement - it does not describe what the manuscript is about.
- I think the keywords are wrongly chosen. After reading them, I expected a completely different content of the manuscript.
- The purpose of the paper is very general and therefore not very specific. It is worth considering the formulation of hypotheses.
- The aim of the work indicated in the abstract is different from that indicated in the introduction - this should definitely be changed, unified.
- The material/data presented is very readable.
- What is missing from the discussion is a reference to the research that has already been done in this area in the world.
- In conclusion, it is worth pointing out the limitations of conducting the study (there are quite a few, only two case studies were analyzed) and the scalability of the research.
I wish you success in improving your manuscript.
Author Response
We want to thank the reviewer for his/her effort and the valuable feedback provided.
Let us going deeply into the comments provided:
1.- Abstract needs improvement - it does not describe what the manuscript is about.
Thanks for the comment. The abstract was updated according to the given comments.
2. I think the keywords are wrongly chosen. After reading them, I expected a completely different content of the manuscript.
The keywords were updated to be closer to the paper's content.
3.- The purpose of the paper is very general and therefore not very specific. It is worth considering the formulation of hypotheses.
We agree with the idea of providing evidences about benefits of seamless integration of different sources of variation to explain process variability, considering single and cross influencing effects. In order to make clear such goal, and following the provided recomendation a research question was introduced in the Introduction section.
4.- The aim of the work indicated in the abstract is different from that indicated in the introduction - this should definitely be changed, unified.
The abstract and the introduction have been aligned to avoid providing a non unified perspective.
5.- The material/data presented is very readable.
Thank you for the positive comment.
6.- What is missing from the discussion is a reference to the research that has already been done in this area in the world.
Thank you for the suggestion. The state of the art was enlarged to better justify what was already done in the filed and what is the gap in our humble opinion.
7.- In conclusion, it is worth pointing out the limitations of conducting the study (there are quite a few, only two case studies were analyzed) and the scalability of the research.
Thank you again for the recommendation. We have added specific messages in conclusion section, to emphasize these aspects.
Again, many thanks for the support given and for the quality of the constructive comments provided. We do hope you can find the next version of the paper acceptable.
Reviewer 2 Report
The paper aims to explore how implementation of IIoT technology and advanced data processing can promote integration strategies, improving I4.0 planning and management. Reflecting on two case studies, where manual and automated processes complements, the authors provide evidences how data analysis can significantly improve process performance understanding. These approaches can be extrapolated in different I4.0 scenarios and management implications. The paper is well written and easy to follow.
Author Response
We want to thank the positive comments provided by the reviewer, as well as the time s/he has invested in reviewing our work.
We have introduced some clarifications in the main text just to address the minor revision required by other reviewers. Therefore, we hope you still consider the contribution helpful.
Reviewer 3 Report
This is a well-written paper and it addresses important aspects of H-M integration.
However, even though the research methodology is well presented, I question the usefulness of two diverse case studies as sufficient to prove or even better understanding the human operator influence in process variability.
Further, the first case study is referring to a loading-unloading truck and a driven which definitely cannot be characterized as a cyber-physical system since the vehicle does not exhibit a "smart behavior". Generally, a CPS is composed of physical and software components deeply intertwined, able to operate on different spatial and temporal scales, exhibiting dynamic behavior and not a simple moving track. Further, the data collected are limited.
The second case study is more suitable for the objectives of the paper.
Generally, these two case studies do not allow the researcher to generalize. As such the findings and the usefulness of this study is limited. Process variability normally is studied on specific processes and proving or extracting general conclusions on how human operators affect process behavior is not feasible.
However, this study can be considered as a first step in the correct direction. Clearly, more work is needed for identifying patterns and behaviors.
Author Response
We want to thank the reviewer for his/her effort and the valuable feedback provided.
Let us going deeply into the comments provided:
1.- This is a well-written paper and it addresses important aspects of H-M integration.
Thanks for the general comment.
2.- However, even though the research methodology is well presented, I question the usefulness of two diverse case studies as sufficient to prove or even better understanding the human operator influence in process variability.
Thank you for your question. We have introduced a research question "RQ1: The I4.0 technology allows to increase the transparency to understand process variability when it is used to integrate different sources of uncertainty." and, according to it the value of the diverse case studies can be better understood. The goal is not to find the same outcomes, but to verify to what end the process variability can be better explained when we integrate different and relevant sources of variability.
3.- Further, the first case study is referring to a loading-unloading truck and a driven which definitely cannot be characterized as a cyber-physical system since the vehicle does not exhibit a "smart behavior". Generally, a CPS is composed of physical and software components deeply intertwined, able to operate on different spatial and temporal scales, exhibiting dynamic behavior and not a simple moving track. Further, the data collected are limited.
Thank you for the comment. Although the reviewer is right, we do believe that the collecting system can be considered 'smart' as we have added different devices providing advanced information for the reverse collecting process. It does include human centered sensors for vehicle drivers, as well as position and route performance, but also indoor climate conditions, and process efficiency by using NFC solutions. All the information is focused at the first stage in understanding process variation helping to implement improving measures, but after understanding reasons for variability, deeply analysis and actions can be addressed.
Although the data collection period can be further extended, it was found enough to explain better sources for process variability than before.
4.- The second case study is more suitable for the objectives of the paper.
Thank you for your comment. We consider it just another different example with different complexity level, but also helping to evaluate the hipothesys
5.- Generally, these two case studies do not allow the researcher to generalize. As such the findings and the usefulness of this study is limited. Process variability normally is studied on specific processes and proving or extracting general conclusions on how human operators affect process behavior is not feasible.
As already explained, although it is true that two case studies become a limitation for very general conclusion, we can also learn that integrating data from different flows which are relevant in terms of variability, and by using IoT and AI technologies, the knowledge level increases getting benefit form the systematic integrated perspective obtained from the process
6.- However, this study can be considered as a first step in the correct direction. Clearly, more work is needed for identifying patterns and behaviors.
Although we agree with the reviewer that this first work can be enlarged and enriched with different cases, it is true that these managerial questions illustrated in these cases are not the end of the game. After bringing better explanation for variabilities, additional managerial questions follow as a natural effect. Therefore, the same principle can be further applied, not only for additional cases, but also by the same cases with additional questions.
Several comments have been added into the paper, looking to clarify all these aspects, in the way the reviewer was suggesting. We do hope such comments can be found convenient.
Round 2
Reviewer 1 Report
The authors have considered all my comments, I recommend the manuscript for publication.